# Conversion of Carbon Dioxide into Methanol Using Cu–Zn Nanostructured Materials as Catalysts

**DOI:** 10.3390/nano12060999

**Published:** 2022-03-18

**Authors:** Anna Carrasco García, Javier Moral-Vico, Ahmad Abo Markeb, Antoni Sánchez

**Affiliations:** 1Departament of Chemical, Biological and Environmental Engineering, Escola d’Enginyeria, Universitat Autònoma de Barcelona, 08193 Cerdanyola del Vallès, Barcelona, Spain; anna.carrasco@uab.cat (A.C.G.); a_markeb@aun.edu.eg (A.A.M.); antoni.sanchez@uab.cat (A.S.); 2Departament of Chemistry, Faculty of Science, Assiut University, Assiut 71516, Egypt

**Keywords:** carbon dioxide hydrogenation, methanol, nanomaterials, heterogeneous catalysis

## Abstract

Nowadays, there is a growing awareness of the great environmental impact caused by the enormous amounts of carbon dioxide emitted. Several alternatives exist to solve this problem, and one of them is the hydrogenation of carbon dioxide into methanol by using nanomaterials as catalysts. The aim of this alternative is to produce a value-added chemical, such as methanol, which is a cheaply available feedstock. The development of improved materials for this conversion reaction and a deeper study of the existing ones are important for obtaining higher efficiencies in terms of yield, conversion, and methanol selectivity, in addition to allowing milder reaction conditions in terms of pressure and temperature. In this work, the performance of copper, zinc, and zinc oxide nanoparticles in supported and unsupported bimetallic systems is evaluated in order to establish a comparison among the different materials according to their efficiency. For that, a packed bed reactor operating with a continuous gas flow is used. The obtained results indicate that the use of bimetallic systems combined with porous supports, such as zeolite and activated carbon, is beneficial, thus improving the performance of unsupported materials by four times.

## 1. Introduction

Climate change stands as one of the greatest issues and challenges that humanity faces, and emissions of greenhouse gases produced by the use of fossil fuels for energy production are identified as the main cause of global warming. Gases such as carbon dioxide and methane are the most important representatives. The alarming increase in atmospheric carbon dioxide concentration caused by anthropogenic emissions has become a serious threat to the planet, causing global warming [1]. During 2018, 33 gigatonnes of carbon dioxide emitted into the atmosphere resulted in an increase in its concentration from 280 to 410 ppm [2]. To tackle this issue, two strategies have been proposed: carbon capture and storage (CCS), whose main drawback is economical justifiability, since it is a non-profitable option [3], and carbon capture and utilisation (CCU) [2], which intends to transform waste carbon dioxide into useful products, such as chemical feedstocks and renewable fuels. There currently exist several technologies that achieve the conversion of carbon dioxide into other value-added products, such as carbon dioxide hydrogenation, electrochemical reduction, photochemical reduction, and photoelectrochemical reduction. Specifically, this project focuses on the catalytic hydrogenation of carbon dioxide, since it is considered as the most commonly used and simplest process [4]. Several chemical products can be obtained from carbon dioxide hydrogenation, among which methanol stands out for two main reasons: its use as a renewable energy source and its utility as a feedstock for obtaining other chemical products. As an energy source, methanol presents the advantage of being liquid at environmental temperature, which makes it easier to transport and store. Moreover, 85% of methanol is used in the chemical industry as a solvent or as a reagent for the synthesis of formaldehyde and acetic acid [4], which are valuable products in industrial terms. In the early 1920s, BASF successfully produced methanol from syngas during the ammonia synthesis process using the ZnO-Cr_2_O_3_ catalyst at a temperature of 320–450 °C and a pressure of 250–350 bar [5]. In the 1960s, ICI (Imperial Chemical Industries) developed a route for methanol synthesis in which syngas containing a high proportion of carbon dioxide reacted with highly selective copper and zinc oxide catalysts, resulting in the first commercially developed Cu/ZnO/Al_2_O_3_ catalyst [6]. This material is currently the most commonly used for the conversion of carbon dioxide into methanol, which means that heterogeneous catalysts are preferable to homogeneous systems due to their high stability and easy use. 

The hydrogenation of carbon dioxide into methanol consists of two main competing reactions. The first is the synthesis of methanol from carbon dioxide and hydrogen [3].
(1)$${\mathrm{CO}}_{2}+3H_{2}\;\to {\mathrm{CH}}_{3}\mathrm{OH}+{\;H}_{2}O\;\Delta H^{0}{}_{298\;K}=-49.5\;\mathrm{kJ}/\mathrm{mol}$$

The second reaction is the reverse water–gas shift reaction (RWGS), which produces carbon monoxide:(2)$${\mathrm{CO}}_{2}+{\;H}_{2}\to \mathrm{CO}+H_{2}O\;\Delta H^{0}{}_{298\;K}=41.2\;\mathrm{kJ}/\mathrm{mol}$$

Furthermore, catalytic hydrogenation of carbon dioxide into methanol can also occur indirectly from carbon monoxide formed through the RWGS reaction:(3)$$\mathrm{CO}+2H_{2}\to {\mathrm{CH}}_{3}\mathrm{OH}\;\Delta H^{0}{}_{298\;K}=-90.6\;\mathrm{kJ}/\mathrm{mol}$$

It is observed in Equations (1) and (3) that, according to Le Chatelier’s principle, the increase in the pressures and the decrease in the temperatures will shift the reaction towards the products to produce more methanol and not towards the production of carbon monoxide. Therefore, catalytic hydrogenation of carbon dioxide into methanol can occur directly or indirectly. 

Among the 92 natural elements present on Earth, copper is 25th most abundant element, which makes the use of copper in catalysis sustainable and economically favourable. For the synthesis of methanol through the hydrogenation of carbon monoxide and carbon dioxide, special emphasis is placed on copper metal acting as the active phase and zinc oxide as the active promoter. The high metal dispersion and stability of the commercial Cu/ZnO/Al_2_O_3_ catalyst is due to the unique microstructure achieved after reduction, where the ZnO nanoparticles act as spacers between the Cu^0^ particles, and Al_2_O_3_ enhances the thermal and chemical stability [6]. Thanks to the development of Cu-Zn-based active catalysts, the reaction pressure of methanol synthesis has been substantially reduced to 50–100 bar, while a satisfactory methanol selectivity of 50% can be achieved. Bimetallic catalysts have emerged as an important class of catalyst due to their unique properties that combine those of the constituent metals, permitting a change in the adsorption properties of metal surfaces, leading to improved catalytic yield [3]. Apart from the Cu-Zn-based system, other bimetallic systems, such as Pd-Zn, Pd-Ga [7], Cu-Ni, Ga-Ni [8], and other ones, have also been investigated. It was recently reported that methanol production is more favourable when using a Pd-Zn based catalyst. It was found that the Pd-Zn surface could act as an unusually high kinetic barrier (activation) for the RWGS reaction, thus effectively enhancing methanol production while reducing carbon monoxide levels under low-pressure conditions (20 bar). Furthermore, it was discovered that Ni-Ga system has a better catalytic activity and lower carbon monoxide production compared to the commercial Cu/ZnO/Al_2_O_3_ catalyst [8]. Hence, it was observed how bimetallic catalytic systems could mitigate the issue of working at high pressures and the low selectivity towards methanol with respect to carbon monoxide, and among, them the Cu-Zn bimetallic system presents certain advantages in economic terms and is by far the most used. In addition to high pressure and low selectivity, another issue arises: the low catalytic yield. In order to overcome this issue, several types of supports are used to increase the adsorption capacity of the active surfaces and to increase the surface area, with the final aim of increasing the catalytic activity. 

Capping agents are of utmost importance as stabilisers that inhibit the overgrowth of nanoparticles and prevent their aggregation [9]. The aim of using a capping agent is to obtain the smallest nanoparticles possible, which would lead to the highest area-per-volume rate. In this work, the capping agent used is polyvinylpyrrolidone (PVP) and sodium citrate. Among all the supports that have been investigated, this work is focused on zeolite, activated carbon, and polypyrrole nanotubes. Zeolite is a kind of microporous aluminosilicate with a three-dimensional structure based on SiO_4_ and AlO_4_, and it is widely known for its catalytic and adsorption properties. Activated carbon is a highly porous material with a carbonaceous structure that is micro- and mesoporous, and it is widely used for adsorption purposes. Both materials have a high adsorption capacity for a wide variety of gas molecules, including carbon dioxide [10,11]. Polypyrrole nanotubes consist of carbon nanotubes containing nitrogen at elevated temperatures. The presence of nitrogen atoms in polypyrrole and its carbonised analogues positively affects their catalytic properties when they are used alone or as supports for noble-metal nanoparticles. The electron pair in nitrogen atoms participates in the adsorption/desorption of reactants/products that is needed for an efficient catalytic performance [12]. Although the main characteristic of polypyrrole nanotubes is their high conductive capacity, we intended to study this material because the direct application of bimetallic nanoparticles is difficult due to their high agglomeration tendency because of their nanometric size and the Van der Waals interactions between metals [13]. 

This work will focus on studying the catalytic yield of hydrogenation of carbon dioxide into methanol using a Cu-Zn-based catalyst with and without support. The aim of this work is to find the bimetallic catalyst with the best catalytic yield and selectivity under the mildest possible operating conditions of temperature and pressure.

## 2. Materials and Methods

### 2.1. Materials

Zinc sulfate heptahydrate (ZnSO_4_·7H_2_O) (99.0%), sodium hydroxide (NaOH) (98.0%), acetic acid (99.7%), copper nitrate trihydrate (Cu (NO_3_)_3_·3H_2_O) (99.0%), zinc nitrate hexahydrate (Zn (NO_3_)_2_·6H_2_O) (98.0%), trisodium citrate (99.0%), copper sulfate pentahydrate (CuSO_4_·5H_2_O) (98.9%), ammonium hydroxide (NH_4_OH) (28.0%), pyrrole (99.0%), iron (III) chloride hexahydrate (FeCl_3_·6H_2_O) (98.0%), methyl orange (85.0%), ethanol (96.0%), sodium borohydride (NaBH_4_) (99.0%), potassium hydroxide (85.0%), 4-Bromo butyric acid (98.0%), dimethylformamide (99.8%), zeolite, activated carbon, and polyvinylpyrrolidone (PVP) were all purchased from Sigma-Aldrich (Barcelona, Spain). A mixed-gas bottle of carbon dioxide and hydrogen with a molar ratio of 1:3, respectively, was provided by Carburos Metálicos, S.A. (Barcelona, Spain). 

### 2.2. Synthesis of Nanocomposites

#### 2.2.1. ZnO Nanoparticle Synthesis

ZnO nanoparticles were synthesised using the co-precipitation method [14]. The co-precipitation of the salts occurs by the addition of a precipitating agent, such as sodium hydroxide [15]. PVP is used to control the size of the nanoparticles and to prevent their agglomeration [9]. Briefly, in 120 mL of deionised water, the precursor salt, 7 g of zinc sulfate heptahydrate at a concentration of 0.2 M, was dissolved, and in another 120 mL of deionised water, the precipitating agent, NaOH, was dissolved at a concentration of 4 M. The synthesis was performed in a 500 mL Scharlau Minireactor HME-R/500 with mechanical stirring and heating. The 240 mL zinc sulfate heptahydrate solution in deionised water was added to the reactor. Then, the NaOH solution was added at a flow rate of 5 mL/min using a peristaltic pump (Watson Marlow SCI 400), and the reaction was carried out at 60 °C for 2 h with constant stirring at 120 rpm [16]. The product was centrifuged (Beckman Avanti J-20 Centrifuge) three times for 15 min at 5000 rpm to remove any impurities and dried in an oven at 105 °C overnight. The synthesis of ZnO with PVP was performed with the same method, but 2 g of PVP was added to the precursor salt solution.

#### 2.2.2. Cu Nanoparticle Synthesis

Cu nanoparticles were synthesised using the chemical reduction method [17]. A total of 10 g of copper sulfate pentahydrate was dissolved in an Erlenmeyer flask with 100 mL of deionised water with magnetic stirring and nitrogen bubbling to avoid the oxidation of copper. After 30 min stirring, the reducing agent, sodium borohydride, was added. A total of 7.5 g of sodium borohydride was dissolved in 50 mL of deionised water and then added with a Pasteur pipette. Once all of the reducing agent was added, the solution was left to stir overnight to ensure that all of the copper had been reduced. Once the reaction was complete, it was centrifuged 3 times at 5000 rpm for 15 min. Finally, it was dried at 105 °C overnight.

#### 2.2.3. CuO/ZnO Bimetallic Nanoparticle Synthesis

CuO/ZnO nanoparticles were synthesised using the co-precipitation method. In 100 mL of deionised water containing 1% *v*/*v* of acetic acid at room temperature, 6 g of copper nitrate trihydrate and 3 g of zinc nitrate hexahydrate were added, and then the pH was adjusted to a value of 7 by the addition of aqueous 1.0 M NH_4_OH. Afterwards, the synthesis of these nanoparticles was carried out in a Scharlau Minireactor HME-R/500 with mechanical stirring, and the temperature was controlled at 60 °C. The product was obtained after centrifugation and drying [18].

#### 2.2.4. Cu/ZnO Bimetallic Nanoparticle Synthesis

Cu/ZnO nanoparticles were synthesised using the chemical reduction method. In 100 mL of deionised water, 1.4 g of previously synthesised ZnO was dispersed in an Erlenmeyer flask. Then, 0.01 g of sodium citrate was added and dissolved for 30 min with magnetic stirring and another 30 min in an ultrasonic bath (Ultrasons P. Selecta. 3000513) to complete the homogenisation of the mixture. When all of the ZnO was dissolved, nitrogen was bubbled into the solution for 10 min. In a beaker, 10 g of copper sulfate pentahydrate was dissolved in 100 mL of deionised water, and it was added to the ZnO solution. The solution was magnetically stirred for 1 h with nitrogen bubbling. After this, the solution was treated with the reducing agent sodium borohydride, of which 7.5 g was dissolved in 50 mL of deionised water that was previously cooled in a refrigerator at 4 °C. Nitrogen bubbling was stopped and, using a Pasteur pipette, the sodium borohydride solution was added dropwise, and the copper was allowed to reduce overnight. The next day, it was centrifuged at 5000 rpm and dried at 105 °C [19]. The synthesis of Cu/ZnO with PVP was performed by the same procedure, but instead of adding sodium citrate, 2 g of PVP was added.

#### 2.2.5. Cu/ZnO/Zeolite Nanocomposite Synthesis

The method for the Cu/ZnO/Zeolite (Cu/ZnO/Z) preparation consisted of first synthesising the nanoparticles (Cu/ZnO) and then immobilising them on zeolite. First, 6 g of zeolite was dispersed in 100 mL of deionised water and 1.4 g of previously synthesised ZnO was dispersed in 100 mL of deionised water. Both solutions were homogenised in an ultrasonic bath for 30 min. The ZnO solution was transferred to the zeolite solution and homogenised again for 30 min. Finally, the same procedure was followed for the Cu/ZnO synthesis [19].

#### 2.2.6. Cu/ZnO/Polypyrrole Nanotube Synthesis

The synthesis of polypyrrole nanotubes with Cu/ZnO nanoparticles (Cu/ZnO/PN) was based on the previous synthesis of polypyrrole nanotubes followed by their corresponding surface functionalisation with carboxylic acid groups and cation exchange with copper ions, followed by the reduction of the metal ions by a reducing agent. First, polypyrrole nanotubes were synthesised. This synthesis was based on the oxidation of pyrrole from a solution of iron (III) chloride in the presence of methyl orange. A total of 1.46 g of iron (III) chloride hexahydrate was dissolved in 90 mL of deionised water, and 0.29 g of methyl orange was dissolved in 90 mL of deionised water in another beaker. The iron (III) chloride solution was added dropwise to the methyl orange solution and allowed to stir at 150 rpm for 10 min. Then, 0.75 mL of pyrrole that was previously distilled using a rotary evaporator (Heidolph Hei-VAP Advantage) was added and left to stir at 150 rpm and at room temperature overnight. The day after, it was filtered (Albet LabScience) under vacuum using water first and then ethanol several times until the filtrate was colourless. The polypyrrole nanotubes were dried at 80 °C for 24 h [13]. Then, the polypyrrole nanotubes were functionalised. A mixture containing 0.4 g of polypyrrole nanotubes, 3.15 g of potassium hydroxide, and 1.77 mL of 4-bromobutyric acid was dispersed in 400 mL of dimethylformamide and stirred at 120 rpm and at 50 °C overnight. The product was washed with water and ethanol to remove potassium hydroxide and unreacted 4-bromobutyric acid, then dried at 80 °C overnight. Once the polypyrrole nanotubes were functionalised, ZnO and Cu nanoparticles were added. For this purpose, ZnO nanoparticles were added first, followed by the copper and the sodium borohydride. In 250 mL of deionised water, 0.3 g of the functionalised polypyrrole nanotubes, 1.5 g of zinc nitrate hexahydrate, and 2 g of sodium hydroxide were mixed. This was stirred at room temperature for 2 h. Next, the mixture was centrifuged 3 times at 5000 rpm for 15 min and dried at 105 °C overnight. Finally, the copper nanoparticles were added to the polypyrrole nanotubes with the ZnO nanoparticles by chemical reduction, wherein 0.3 g of polypyrrole nanotubes with ZnO, 2.5 g of PVP were dissolved in 150 mL of deionised water. Next, 6.5 g of copper nitrate trihydrate was added and stirred for 2 h. A total of 5.7 g of sodium borohydride was dissolved in 50 mL of deionised water and added afterwards to the solution dropwise with a Pasteur pipette. It was left to stir at 150 rpm and at room temperature overnight, centrifuged 4 times at 5000 rpm for 15 min, and dried in an oven at 105 °C overnight [13].

#### 2.2.7. Cu/ZnO/Activated Carbon Synthesis

The synthesis of Cu/ZnO/activated carbon (Cu/ZnO/AC) was carried out using the impregnation method, which was based on the prior synthesis of Cu/ZnO nanoparticles followed by their adsorption into the support. First, the Cu/ZnO synthesis was performed, and the nanoparticles were dispersed in water. Then, 2.9 g of activated carbon was added and stirred for 3 h. The mixture was centrifugated 5 times and dried [20].

### 2.3. Characterisation of Catalysts

X-ray diffraction (XRD) was used to perform a structural analysis of the nanoparticles and their crystallographic study [21]. The X-ray diffraction patterns were recorded in a diffractometer (PANalytical X’Pert) using Cu–Kα radiation. The measurements were made at room temperature at a range of 3.5°–85.0° on 2θ with a step size of 0.0167°. All XRD results were analysed with the simulation of the nanoparticles obtained using the Mercury software provided by the Cambridge Crystallographic Data Centre (CCDC), who developed the Mercury software, which provides the full Cambridge Structural Database (CSD). Mercury offers a comprehensive range of tools for 3D structure visualisation and the exploration of crystal packing. This software is able to load structural data from a variety of formats and provides an extensive array of options to aid the investigation and analysis of crystal structures [22]. A scanning electron microscopy (SEM) (FEI Quanta 650F ESEM) equipped with an energy-dispersive spectroscopy (EDS) source was used to determine the morphology, size distribution, and composition of the nanoparticles. Transmission electron microscopy (TEM) (FEI TECNAI G2 F20) was used to determine the microstructure of the samples and to analyse the size, morphology, and size distribution of the nanoparticles. Samples were prepared on copper and graphite grids (TED PELLA, Inc., Redding, CA, USA).

### 2.4. Catalytic Activity Test

The catalytic activity of the hydrogenation of carbon dioxide into methanol was evaluated in a stainless-steel-packed bed reactor with an internal diameter of 5.25 mm, a height of 8.90 cm, and, thus, a volume of 1.92 cm^3^. All samples were first calcinated at 350 °C for 4 h, and those containing elemental copper were reduced by a hydrogen flow of 40 mL/min at 350 °C for 2 h in order to ensure its reduction. A certain amount of catalyst was added and a small amount of glass wool was added at each end of the reactor to avoid possible leakage of the material itself. In order to investigate the impact of pressure, the work was carried out under two moderate pressure values: 10 and 15 bar. The flow rate of the stoichiometric H_2_/CO_2_ mixture was 10 mL/min, and the tested temperatures were 180, 200, 220, 240, 260, and 280 °C. The samples were collected using sampling bags (SKC FlexFoil PLUS Sample Bag), and the obtained methanol was analysed on a chromatograph (Shimadzu GC-2010) with a flame ionisation detector (FID) with helium as the carrier gas. The software used was Chromeleon, the inlet temperature was 260 °C, and the flow was 50 mL/min; the detector temperature was 280 °C and the flow was 35 mL/min. An Agilent 7890B GC System chromatograph served to detect carbon monoxide and dioxide by using a thermal conductivity detector (TCD) with helium as the carrier gas. The software used was OpenLab, the inlet temperature was 120 °C, and the inlet flow was 20 mL/min; the detector temperature was 150 °C, and the detector flow was 50 mL/min. For the study of the catalytic activity of each material, three parameters were calculated: space–time yield (STY), which indicates the mass of methanol obtained per mass of catalyst and time, conversion (%), and methanol selectivity, measured as the percentage of carbon dioxide that was converted into methanol rather than carbon monoxide. In the selectivity calculation, it was assumed that only methanol or carbon monoxide was produced. 

## 3. Results and Discussion

### 3.1. Structural and Morphological Characterisation of Nanomaterials

The XRD patterns obtained from monometallic nanomaterials and their corresponding simulations using the Mercury software are presented. In Figure 1, it is shown that the highest peaks of copper correspond to the diffraction angles of 43.4°, 50.5°, and 74.5°, which correspond to the (111), (200), and (220) planes. These highest peaks were also found in the literature [23]. Furthermore, it was observed that the obtained patterns for synthesised copper materials fit those obtained using the simulation. Regarding the zinc oxide with and without PVP, it was observed that PVP did not interfere with the crystallinity of zinc oxide, and the same peaks as in the simulation were obtained at all corresponding diffraction angles. The 2θ angles and their corresponding crystallographic planes were the following: 31.5° (100), 34.5° (002), 36.3° (101), 47.6° (102), 56.9° (110), 63.3° (103), and 68.0° (112). These highest peaks were also found in the literature [18]. 

The X-ray diffraction patterns of the bimetallic nanomaterials without support are shown in Figure 2. The diffraction peaks obtained for CuO/ZnO and their CuO simulation were at the same angles: 32.5° (110), 35.5° (002), 38.8° (111), 48.5° (202), and 61.5° (113). These highest peaks were also found in the literature [18]. However, it was observed that Cu/ZnO had two diffraction peaks at the angles of 35.6° (002) and 39.1° (111), which correspond to CuO. This indicates that the copper was not completely reduced. Small diffraction peaks were seen at the angles of 47.6°, 63.3°, and 68.0°, corresponding to zinc oxide. Hence, it can be suggested that the diffraction peaks obtained were very small due to the low amount of zinc oxide in the material. 

Figure 3 depicts the XRD patterns for bimetallic catalysts with support. First of all, the Cu/ZnO/Zeolite nanocomposite was analysed, observing that the diffraction peaks from 5° to 30° correspond to zeolite [24]. There were also two peaks that correspond to zinc oxide: 32.0° and 34.6° [E]. Regarding the two remaining composite catalysts, the Cu/ZnO/activated carbon and Cu/ZnO/polypyrrole nanotubes, it was observed that both materials presented the same XRD patterns with two large peaks at 35.7° and 38.9°, which correspond to copper oxide [18]. In addition, small peaks corresponding to zinc oxide were observed at 56.9° and 68.0° [18]. Polypyrrole nanotubes and activated carbon did not present XRD patterns due to their amorphous microstructures.

The presence of copper oxide in some materials meant that not all of the copper was reduced. It is assumed to be for the following two reasons: The oxygen present in the solution was not completely displaced, or it could be due to the amount of the reducing agent not being enough. 

The morphology of the materials was examined using a scanning electron microscope (SEM) equipped with EDX. The morphology of the three supports used in this work is shown in Figure 4. The polypyrrole nanotubes (Figure 4A) presented an approximate length of 3–5 μm and a diameter of 0.3–0.6 μm [25]. Furthermore, it was observed that all of the tubes had similar sizes. Regarding the activated carbon (Figure 4B), it was observed that there were small and large granules, probably because the material was not ground and sifted. Zeolite had a square morphology (Figure 4C), and the sizes were homogeneous, as expected [26]. 

Figure 5A depicts the granular morphology of the copper nanoparticles with homogeneous dimensions and distribution. The diameter of the nanoparticles was between 0.27–0.40 μm, as in the literature [23]. Zinc oxide synthesised with and without PVP is shown in Figure 5B–C, and it was observed that when the PVP was added, the nanoparticles obtained were smaller and had better distribution. Moreover, the geometry of the nanoparticles was more defined due to the presence of PVP, as observed in Figure 5B, in which the morphology of the nanoparticles obtained presented a needle shape with larger dimensions than those of the ZnO nanoparticles with PVP (Figure 5C).

Figure 5D depicts the granular morphology of the CuO/ZnO nanoparticles with homogeneous dimensions and distribution. The diameter of the nanoparticles was between 0.13–0.26 μm, as in the literature [18]. As for the bimetallic nanomaterials, the Cu/ZnO nanoparticles were granular with no observed agglomeration. The comparison of the use of PVP in the bimetallic catalyst was also performed for ZnO. In this case, the morphology obtained during the synthesis when using PVP was not well defined, but it presented a tendency towards agglomeration instead. This may have occurred because the copper nanoparticles were wrapped in PVP, which reduced their interaction with ZnO. Nevertheless, when PVP was not added, it was observed how nanoparticles were granular in some areas, and furthermore, the interaction between both metals was stronger. 

Regarding the images of the materials with supports, the Cu/ZnO nanoparticles–polypyrrole nanotubes are shown in Figure 5G, with the nanotubes indicated by an arrow. It was observed that incorporating a support had the advantage of reducing the agglomeration and improving the distribution of the nanoparticles. Figure 5H shows the successful impregnation of Cu/ZnO nanoparticles on the porous structure of activated carbon. The circumferences indicate the activated carbon and the triangle indicates the bimetallic nanoparticles. The enhancement of the nanomaterials in terms of nanoparticle aggregation and distribution when a support was used was finally confirmed by Figure 5I, in which the nanocomposite of Cu/ZnO/zeolite is shown. This was translated into an increased specific surface, which means that the active sites of the catalyst were more exposed, increasing the catalytic activity. 

The composition of the chemical elements present in the catalysts was analysed using energy-dispersive X-ray spectroscopy (EDX). This analysis was performed on the two samples with porous supports: zeolite and activated carbon. It was determined that the catalyst with zeolite (Table 1) had the expected chemical elements, the two metals of the nanoparticles (Cu and Zn), and the two typical elements of zeolite (Al and Si). Oxygen was present in both the nanoparticles and the support. Regarding the EDX data of the activated carbon (Table 2), all present elements were those that were expected. There was a non-significant amount of sodium remaining due to the use of sodium borohydride during the synthesis of the material.

To clarify some of the data obtained regarding the catalytic efficiency of the synthesised materials, transmission electron microscopy (TEM) was used. The first sample analysed was Cu/ZnO with PVP due to its poor results in terms of catalysis, despite the addition of a stabilising agent. As expected, in the Figure 6A some aggregates of these nanoparticles were observed, and an unsatisfactory homogenisation of the material was seen. The Cu/ZnO with PVP nanoparticles obtained in this synthesis was bigger compared to that in the other images. As mentioned above, SEM images did not distinguish the nanoparticles and the support well in the Cu/ZnO samples with polypyrrole nanotubes and zeolite. Therefore, the TEM images will provide more information about the distribution of nanoparticles and the support. The Cu/ZnO/PN was the second catalyst to confirm whether the synthesis was successful and whether the nanotubes fulfilled their main function of avoiding the agglomeration of these nanoparticles. Figure 6B,C depict that there was no agglomeration indeed, and it was observed that they were smaller than those obtained in the previous catalysts. The last sample was Cu/ZnO/zeolite, which was the catalyst with the best performance. In this sample, the distribution of the chemical compounds was homogeneous, and the size of these nanoparticles was smaller, giving rise to an increased specific surface, which fit the catalysis results.

### 3.2. Catalytic Activity of Nanomaterials 

The space–time yield (STY) evaluated the amount of methanol produced per mass unit of catalyst and time. It was determined for each catalyst used at different temperatures, as shown in Figure 7. In general, the production of methanol increased with temperature. Nevertheless, for some materials, it was observed that the same values were obtained for the STY at 260 and at 280 °C; even in the case of Cu/ZnO/PN, a decrease in the STY was observed in this range. This could be related to the selectivity of the reaction, which can favour the production of carbon monoxide compared to methanol at these high temperatures. If we consider the results of the STY analysis when classified by types of materials, monometallic materials should be the first to be analysed. In particular, the results indicate that methanol was hardly obtained with copper—only at high temperatures and with non-significant concentrations of methanol. The lower activity of the copper was suggested to result from formate poisoning of the copper surface, which is an intermediate of the hydrogenation of carbon dioxide into methanol [27]. Furthermore, copper was found to be ineffective as a catalyst material for its low surface area [28]. Regarding zinc oxide, the methanol concentrations were slightly higher than those of copper. However, these concentrations were small compared with those of the other catalysts. The difference in methanol concentrations between zinc oxide and copper and the role that these two metals played in methanol synthesis was previously analysed in the literature, where it was discussed that zinc oxide is fundamental for stabilising reaction intermediates and reducing energy barriers for the production of methanol [28]. For these two reasons, more methanol was produced with zinc oxide than with copper. Zinc oxide synthesised with and without PVP was compared in terms of methanol STY, with better results for the case synthesised with PVP. 

As observed in Section 3.1 (Figure 5B,C), adding PVP entailed the obtention of smaller nanoparticles with more regular morphology, which made the specific surface higher. The bimetallic materials were compared as well. It was observed that with CuO/ZnO, the amount of methanol produced was lower than with Cu/ZnO. It was found in the literature that the active component is Cu^0^, and although Cu^2+^ can also act as an active component, it offers less catalytic activity [29,30].

Moreover, several publications reported that copper nanoparticles are an active phase for carbon dioxide hydrogenation, and zinc oxide helps to disperse these copper nanoparticles on the surface of the catalyst [31,32]. Thus, the presence of zinc oxide increases the exhibition of copper active sites, as it is a physical spacer between these nanoparticles. Nevertheless, the most recent literature suggests that Zn atoms at the edges of copper nanoparticles are also active sites [33]. Thus, metallic Zn on the copper surface increases the binding energy of oxygen-bound intermediates, that is, formate, and this facilitates the hydrogenation pathway of carbon dioxide. Therefore, zinc oxide also catalyses the reaction, although in smaller quantities than Cu/ZnO because the active sites for carbon dioxide hydrogenation are found in both metals.

A complete study was performed on Cu/ZnO, and the effect of adding PVP was also compared. In the case of this material, the results clearly showed that adding PVP to a bimetallic system is not a viable alternative. The possible advantages of the use of supports in the production of methanol were analysed. For this purpose, the catalyst Cu/ZnO without PVP was chosen due to its good results, and three types of supports were chosen. The first supports tested were polypyrrole nanotubes in order to solve the problem of nanoparticle agglomeration and, thus, to increase the exposure of active sites. Figure 7 shows the obvious difference between Cu/ZnO and Cu/ZnO with polypyrrole nanotubes, confirming the effect of the support in terms of methanol production. The other two supports used were zeolite and activated carbon, which are both porous materials. The best results of conversion from carbon dioxide to methanol were achieved with the use of these porous materials, thanks to the synergy between the carbon dioxide adsorption capacity of the supports and the active sites of the Cu/ZnO catalyst. Nevertheless, the Cu/ZnO/zeolite material produced the best results, probably due to zeolite’s higher carbon dioxide adsorption capacity per gram of material [10,11]. 

These three supports were analysed to verify whether they catalysed methanol by themselves. Therefore, a catalytic test was performed at the same conditions of temperature, pressure, and flow as with the other catalyst. The results indicated that none of them had catalytic activity.

The methanol selectivity of all of the studied catalysts is shown in Figure 8. In all of the cases, from 180 to 220 °C, the methanol selectivity obtained was 100%, which means that no carbon monoxide was produced at these temperatures; thus, only the methanol selectivity values from 240 to 280 °C are plotted in Figure 8. Almost all catalysts had 100% methanol selectivity at 240 °C, but as the temperature increased, in general, all materials lost selectivity. This was due to the standard enthalpy of the reaction of methanol being negative; hence, it is an exothermic reaction that is favoured at low temperatures. However, in three catalysts, the amount of carbon monoxide obtained was zero for temperatures up to 280 °C. These catalysts were zinc oxide with and without PVP and Cu/ZnO with PVP. As stated in the literature [27], the catalytic activity and methanol selectivity can be correlated with the increase in Zn^0^ concentration. It was discovered that a zinc enrichment in the catalyst leads to greater activity and methanol selectivity due to two factors: improving the Cu and ZnO interactions that increase carbon dioxide hydrogenation and inhibiting the RWGS reaction, which produces carbon monoxide [34]. In the case of materials synthesised using PVP, this lack of carbon monoxide was possibly due to the poor distribution of zinc oxide and copper nanoparticles, which were also agglomerated. As expected, the active sites of copper were poisoned by formate, and only the active sites of zinc oxide were available. 

The relationships between the methanol selectivity and the CO_2_ conversion for some catalysts with good STY results are shown in Figure 9. The displayed values correspond to the methanol selectivity and conversion of carbon dioxide from 240 to 280 °C. As observed, the general behaviour was that as selectivity decreased, conversion and temperature increased. Figure 9 shows that the catalyst with the highest carbon dioxide conversion was Cu/ZnO/Zeolite, similarly to the STY results seen in Figure 7. The yield and conversion data were low compared with the literature. For example, using Cu/ZnO plates, the maximum CO_2_ conversion reported was 8% [35,36]. However, the operating conditions of pressure in these studies were higher (30 bar) than those of this work (15 bar). Even though many existing publications studied the catalytic properties of similar materials, it is very rare to find documents with studies operating at pressures lower than 20 bar [37,38].

Finally, the STY values and selectivity towards methanol of all the catalysts at 260 °C and the temperature with the maximum STY values are plotted in Figure 10. 

Figure 10 shows that the catalyst that produced the highest yield of methanol was Cu/ZnO/zeolite. As expected, by adding a support such as zeolite to the Cu/ZnO bimetallic system, the STY values obtained were 76.75% higher than without the support. It can also be stated that bimetallic systems have a better catalytic efficiency than ZnO and Cu^0^ by themselves, with the latter having no methanol production.

## 4. Conclusions

Several types of copper and zinc compounds— in particular, single metals, their oxides, and bimetallic materials, which were embedded or not embedded in several types of supports—were successfully synthesised in this study, and their catalytic efficiency was studied. In general, the bimetallic materials showed a higher efficiency than the single-metal materials in terms of methanol yield. In these types of materials, both metals (Cu and Zn) had active sites and differentiated functions, as zinc acted physically as a spacer, exposing the active sites of both metals on the surface, which significantly helped their performance as catalysts. Even though adding PVP as a capping agent improved the catalytic capacity of single-metal nanoparticles, since it normally helps to achieve more homogeneous and smaller particles, in the case of the bimetallic materials, it hindered the correct homogenisation of both metals, causing an evident tendency to agglomerate, which clearly diminished their catalytic capacity. The great importance of using a support was also confirmed by using both nanotubes for a greater spacing of the active sites or with porous supports such as zeolite or activated carbon, which could improve the distribution of metals and the adsorption of carbon dioxide. These two particular features of porous supports clearly improved the catalytic efficiency of these nanocomposites when compared with the other studied materials. 

## Figures and Tables

**Figure 1 nanomaterials-12-00999-f001:**
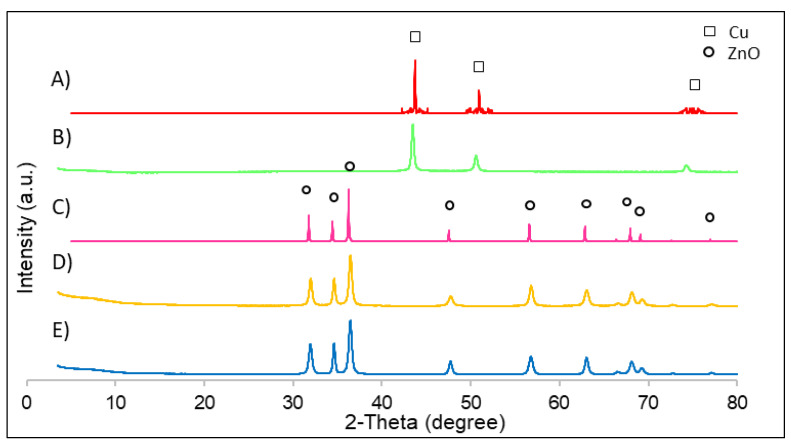
XRD patterns of all catalysts: (**A**) Cu nanoparticle simulation; (**B**) Cu nanoparticles; (**C**) ZnO nanoparticle simulation; (**D**) ZnO nanoparticles with PVP; (**E**) ZnO nanoparticles.

**Figure 2 nanomaterials-12-00999-f002:**
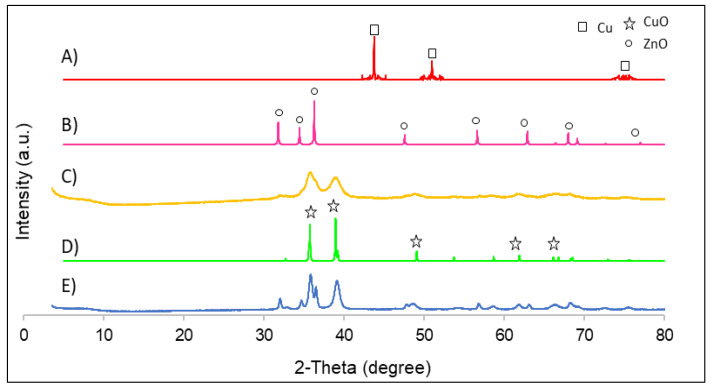
XRD patterns of all catalysts: (**A**) Cu nanoparticle simulation; (**B**) ZnO nanoparticle simulation; (**C**) Cu/ZnO nanoparticles; (**D**) CuO nanoparticle simulation; (**E**) CuO/ZnO nanoparticles.

**Figure 3 nanomaterials-12-00999-f003:**
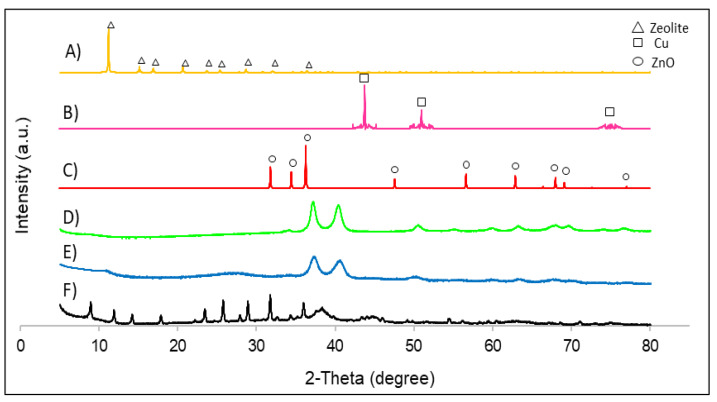
XRD patterns of all catalysts: (**A**) zeolite simulation; (**B**) Cu nanoparticle simulation; (**C**) ZnO nanoparticle simulation; (**D**) Cu/ZnO/polypyrrole nanotubes; (**E**) Cu/ZnO/activated carbon; (**F**) Cu/ZnO/zeolite.

**Figure 4 nanomaterials-12-00999-f004:**
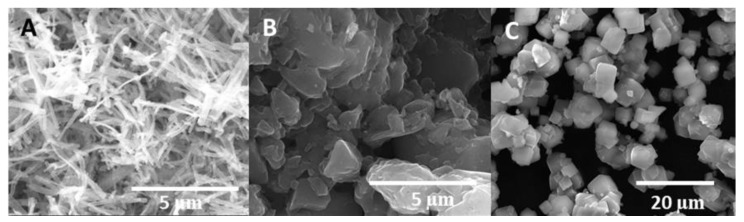
SEM images of the supports: (**A**) polypyrrole nanotubes; (**B**) activated carbon; (**C**) zeolite.

**Figure 5 nanomaterials-12-00999-f005:**
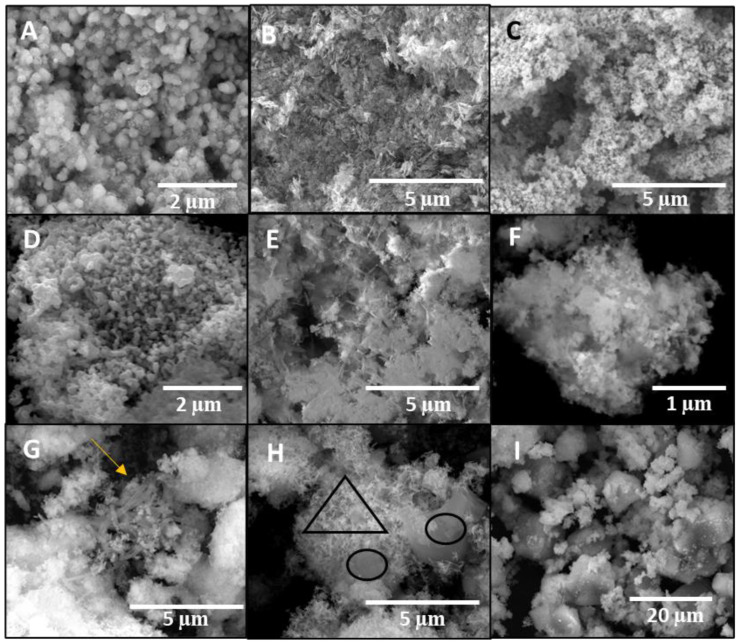
SEM images of the catalysts: (**A**) Cu nanoparticles; (**B**) ZnO nanoparticles; (**C**) ZnO nanoparticles with PVP; (**D**) CuO/ZnO bimetallic nanoparticles; (**E**) Cu/ZnO bimetallic nanoparticles with PVP; (**F**) Cu/ZnO bimetallic nanoparticles; (**G**) Cu/ZnO/polypyrrole nanotubes; (**H**) Cu/ZnO/activated carbon; (**I**) Cu/ZnO/zeolite. The arrow indicates the polypyrrole nanotubes, the triangle indicates the bimetallic nanoparticles, and the circumferences indicate the activated carbon.

**Figure 6 nanomaterials-12-00999-f006:**
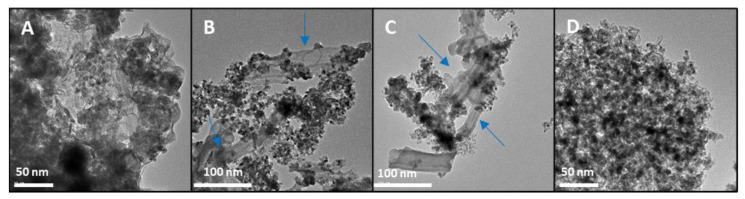
TEM images. (**A**) Cu/ZnO bimetallic nanoparticles with PVP; (**B**,**C**) Cu/ZnO/polypyrrole nanotubes; (**D**) Cu/ZnO/zeolite. The arrows indicate the polypyrrole nanotubes.

**Figure 7 nanomaterials-12-00999-f007:**
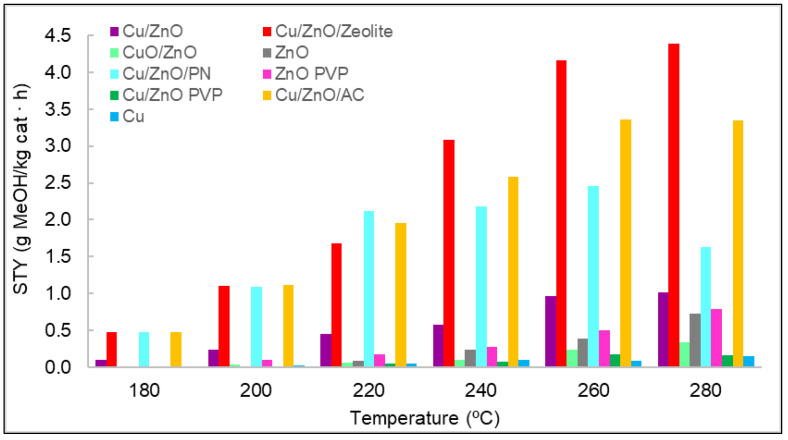
STY values for each catalyst at different temperatures and at a pressure of 15 bar.

**Figure 8 nanomaterials-12-00999-f008:**
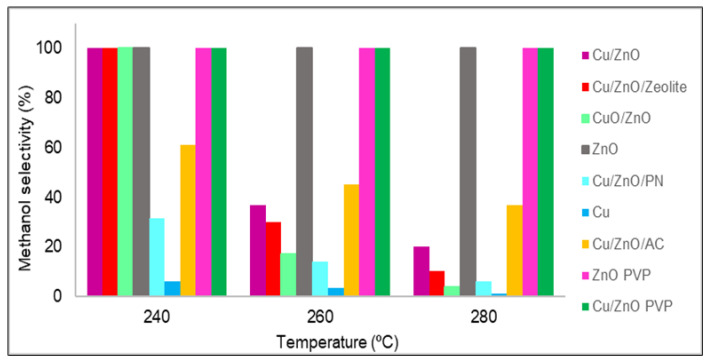
Methanol selectivity (%) values of all catalysts.

**Figure 9 nanomaterials-12-00999-f009:**
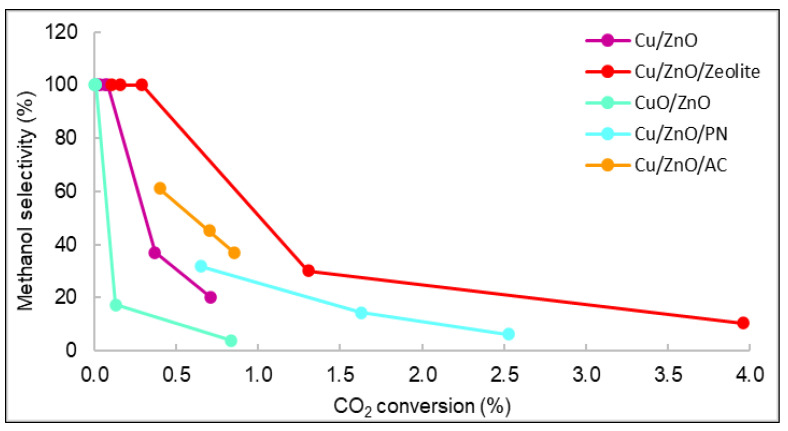
Representation of the selectivity vs. carbon dioxide conversion of all catalysts.

**Figure 10 nanomaterials-12-00999-f010:**
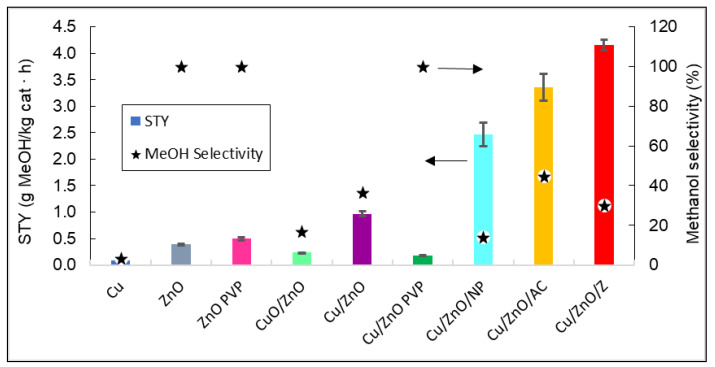
Representation of the STY and the methanol selectivity of all catalysts. The arrows indicate that methanol selectivity values can be read in the right vertical axis, and STY in the left vertical axis.

**Table 1 nanomaterials-12-00999-t001:** Element quantification with EDX for the Cu/ZnO/zeolite material.

Element	Weight (%)	Atomic (%)
O	46.93	70.51
Al	9.67	8.61
Si	9.60	8.21
Cu	21.84	8.26
Zn	11.98	4.40

**Table 2 nanomaterials-12-00999-t002:** Element quantification with EDX for the Cu/ZnO/activated carbon material.

Element	Weight (%)	Atomic (%)
O	20.90	21.05
C	53.94	72.39
Na	0.51	0.36
Cu	16.97	4.31
Zn	7.68	1.89

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
