# Peer review of "Conversion of Carbon Dioxide into Methanol Using Cu–Zn Nanostructured Materials as Catalysts"

_nanomaterials, 2022, doi:10.3390/nano12060999_

Round 1

Reviewer 1 Report

The manuscript deals with the conversion of CO2 to methanol on mainly Cu/ZnO catalyst. There is a vast amount of paper published on the subject. The catalysts were used as nanoparticles and also deposited on zeolite (Aldrich), polypyrrole, and activated carbon. The detailed structural characterizations of the catalysts used have been provided, which might be interesting for the scientific community. However, the reported values of the methanol yields (up to 4 g of methanol per 1 kg (?) of a catalyst in 1 h) and CO2 conversion (up to 4 %) are well below those reported in the literature. In several experiments 100 % methanol selectivity has been achieved, however in the cases the conversion of CO2 was less than 0.5 %.  Nevertheless, taking into account the importance of the subject, I would recommend the manuscript for publication.

However, the text needs to be revised before publication. Some remarks are in the order of appearance in the text:

Lines 158, 202, and 224 - the statement “seven point five grams” instead 7.5 g looks awkward.

Line 180 – “the solution was prepared with”, I would suggest – the solution was treated with.

Line 213 – “The product was filtered with water and methanol…”, The product was washed with water and methanol…,

Line 214 – “4- Bromo butyric”, 4-bromobutyric,

line 372 – “the presence of sodium indicates that it should have been centrifuged more times”, why not check it?

line 400 – “unit of methanol and time”, unit of catalyst and time

lines 419-421 – “Zinc oxide with 419 and without PVP was compared in terms of methanol STY, with better results for the case with PVP presence.”, I am not a native English speaker and I hesitate to make remarks about language, but I do not understand the statement.

line 478 – “…the methanol selectivity and the conversion of some catalysts.” …the methanol selectivity and the CO2 conversion for some catalysts.(?)

Reviewer 2 Report

This study has made an attempt to produce nanomaterials for the conversion of carbon dioxide to methanol such as copper, copper oxide and zinc oxide. It showed that use of copper and zinc nanoparticles in supported and unsupported bimetallic systems can determine the best catalytic material. It was argued that bimetallic materials display higher efficiency than single metal materials in terms of methanol yield. The importance of using a support has also been confirmed by using nanotubes for a higher spacing of the active sites or with porous supports such as zeolite or activated carbon.

  • Abstract and conclusions are not well rewritten and compact. They should be rewritten.
  • 7-10 look good, but the authors have not discussed them intensively.
  • The results presented are not compared with those of similar catalytic systems with similar functionalities that have been reported by several others.
  • I cannot see any Crystal information file in this communication. XRD data needs more discussion, especially when bonding features are discussed.
  • Language used for the ms should be significantly improved, as several parts of the paper are weakly presented.

Reviewer 3 Report

Manuscript ID: nanomaterials-1635117

This is an interesting submission addressing one of the threshold research fronts in Chemistry. The work is systematically performed and the observed results are nicely discussed. Considering the importance of the work and the results presented in this work, I recommend major revision based on the following comments.   

  • The stability of Cu/ZnO/Zeolite should be reported in multiple cycles or under flow conditions. Further, the recycled catalyst must be characterized and be compared with the fresh solid.
  • The activity performance should also be compared with the literature reports summarizing the superior performance of this solid. The activity data should be compared and the unique features of this solid should be highlighted.
  • Mechanistic part is not addressed.
  • Line 109: Activated carbon is a highly crystalline material… Is it true?
  • Line 259: flow rate of what?
  • Lines 379-394: Authors use in many places the word “nanoparticles”. However, it is better to indicate which nanoparticle?
  • XPS analysis is missing. Authors may provide these details to better understand the catalyst structure.
  • Authors should include some key reviews related to this field.
  • Typographical errors have to be checked.

Round 2

Reviewer 2 Report

Authors of this work have made changes in their manuscript, which are highlighted in yellow. I believe that there is an improvement in the quality of the work. I suggest publication of the work. 

Reviewer 3 Report

I am really surprised to see the responses provided by authors for my earlier comments. Authors have responded that most of my concerns were out of the scope of investigation.